# Impact of Rap-Phr system abundance on adaptation of *Bacillus subtilis*

Ramses Gallegos-Monterrosa [1,7], Mathilde Nordgaard Christensen [2,7], Tino Barchewitz[1], Sonja Koppenhöfer[1,6], B. Priyadarshini [2], Balázs Bálint[3], Gergely Maróti[4], Paul J. Kempen [5], Anna Dragoš [2] & Ákos T. Kovács [1,2✉]

Microbes commonly display great genetic plasticity, which has allowed them to colonize all ecological niches on Earth. *Bacillus subtilis* is a soil-dwelling organism that can be isolated from a wide variety of environments. An interesting characteristic of this bacterium is its ability to form biofilms that display complex heterogeneity: individual, clonal cells develop diverse phenotypes in response to different environmental conditions within the biofilm. Here, we scrutinized the impact that the number and variety of the Rap-Phr family of regulators and cell-cell communication modules of *B. subtilis* has on genetic adaptation and evolution. We examine how the Rap family of phosphatase regulators impacts sporulation in diverse niches using a library of single and double *rap-phr* mutants in competition under 4 distinct growth conditions. Using specific DNA barcodes and whole-genome sequencing, population dynamics were followed, revealing the impact of individual Rap phosphatases and arising mutations on the adaptability of *B. subtilis*.

[1] Terrestrial Biofilms Group, Institute of Microbiology, Friedrich-Schiller-University Jena, Jena, Germany. [2] Bacterial Interactions and Evolution Group, DTU Bioengineering, Technical University of Denmark, Kongens Lyngby, Denmark. [3] Seqomics Biotechnology Ltd., Mórahalom, Hungary. [4] Institute of Plant Biology, Biological Research Centre, Hungarian Academy of Sciences, Szeged, Hungary. [5] Department of Health Technology, Technical University of Denmark, Kongens Lyngby, Denmark. [6]Present address: Department of Biology, Memorial University of Newfoundland, St. John's, NL, Canada. [7]These authors contributed equally: Ramses Gallegos-Monterrosa, Mathilde Nordgaard Christensen. ✉email: atkovacs@dtu.dk

In recent years, *Bacillus subtilis* has become a model organism for the study of biofilms and population differentiation due to its ability to develop diverse phenotypes within an isogenic population[1,2]. Even when grown in liquid mixed cultures, where environmental conditions are assumed to be homogeneous, *B. subtilis* cells can be found as single flagellated cells or as non-flagellated chained cells due to stochastic variation in the expression of motility-related genes[3]. This population heterogeneity further increases when *B. subtilis* develops a biofilm, i.e., cells commit to particular functions, such as biofilm matrix production, exoenzyme secretion, or spore formation[1,4]. The development of these different cell types is partially triggered by the variation in the environmental conditions that exist in diverse parts of the biofilms, which can then be seen as a collection of ecological microniches, each with its own type of specialized inhabitant[5–7].

*B. subtilis* possesses a complex regulatory network that leads the cells within the biofilm to generate a phenotypically heterogeneous population. This network is described to be mainly controlled by the master transcriptional regulators Spo0A, DegU, and ComA. The activity of these regulators depends on their phosphorylation status, which is controlled by the activity of specific kinases that can sense a wide array of environmental and intracellular signals, and phosphorylate their corresponding response regulators accordingly. DegU and ComA are phosphorylated by kinases DegS and ComP, respectively, while Spo0A can be activated by five different kinases that act through a phosphorelay formed by the response regulators Spo0F and Spo0B. Furthermore, the regulatory network includes multiple cross-talk mechanisms and regulatory feedback loops that contribute to its modulation by constantly monitoring the general metabolic state of each cell within the biofilm[2,8,9].

The population-heterogeneity regulatory network of *B. subtilis* is further controlled by a family of response regulator aspartyl-phosphate (Rap) phosphatases and their cognate phosphatase-regulator (Phr) peptides. The cytoplasmic Rap proteins exert their regulatory function by inhibiting the activity of their target regulator (Spo0F, DegU, or ComA) via dephosphorylation, or by directly blocking DNA binding. The Rap proteins are, in turn, inhibited by their cognate Phr peptides, which are produced as pre-Phr proteins that are exported to the extracellular milieu and cleaved to produce mature five to six amino acid Phr peptides. The Phr peptides are imported back into the cell upon reaching threshold concentrations at high cell density and bind to their cognate Rap phosphatase, inducing conformational changes that inhibit its activity[10,11]. The *rap* and *phr* genes are usually found as pairs in the same genetic loci, with the *phr* genes following and in some, but not all cases slightly overlapping the corresponding *rap* genes, and therefore the expression of these genes being transcriptionally coupled[12–14]. Furthermore, some of the *rap* genes are not followed by a corresponding *phr* gene. Additionally, some Rap proteins can be regulated by Phr peptides that are encoded in other cassettes, e.g., RapB and RapJ are controlled by PhrC[15].

The Rap–Phr regulatory pairs are highly prevalent in the *Bacillus* genus, with ca. 2700 *rap* genes recently reported to be distributed among 346 *Bacilli* genomes; from those, ca. 80 different putative *rap–phr* alleles were found in the *B. subtilis* group alone[16]. Only a small minority of the *B. subtilis* Rap phosphatases has been characterized, finding that they have high redundancy in their regulatory function: most of them target Spo0F, ComA, or both; and only one (RapG) has been described to act on DegU[11]. Interestingly, *B. subtilis* shows great genomic plasticity regarding *rap–phr* gene pairs; 127 recently compared strains of the *B. subtilis* group were shown to have multiple *rap–phr* gene pairs, with an average of 11 *rap* genes per strain[16]. This genetic variation among *B. subtilis* strains is not superfluous: since the Rap

phosphatases modulate the activity of the main regulators of population heterogeneity, it has been proposed that the Rap–Phr pairs serve to adjust this regulatory network of *B. subtilis* to the needs of particular ecological niches[17,18]. As an example, it has been shown that *B. subtilis* strains isolated from gastrointestinal tracts of diverse animals have diverse sporulation initiation rates, with some being able to start sporulating already during logarithmic growth phase. This variation is correlated to the presence or absence of specific Rap–Phr pairs, and thus it has been suggested that the precise combination of *rap–phr* gene pairs matches the particular sporulation needs of a given niche[17].

Since the Phr peptides function as quorum sensing molecules regulating the production of costly public and private goods[10], bacterial social interactions and evolutionary dynamics have been suggested to influence the emergence and persistence of multiple Rap–Phr systems with redundant functions in *B. subtilis*[15,16]. This idea is supported by the fact that up to 75% of *rap–phr* gene pairs are located in sections of the genome related to mobile genetic elements (such as prophages and transposons), suggesting that Rap–Phr systems are commonly acquired by horizontal gene transfer among *Bacillus* strains[16]; moreover, at least one *B. subtilis* Rap phosphatase (RapI) is known to promote the propagation of the mobile genetic element that contains it[19]. The genetic variation in Rap–Phr systems among *B. subtilis* strains complicates the understanding of the role that the whole set of Rap phosphatases plays in modulating the population-heterogeneity regulatory network of any particular *B. subtilis* strain. Furthermore, the best-known Rap–Phr systems have usually been studied independently from each other, in diverse genetic backgrounds, and using different experimental conditions[10,11]. Likewise, previous investigations have focused on different aspects of Rap–Phr regulation, e.g.: RapA and RapB have been mainly studied for their role in sporulation regulation[20], while RapC and RapF are known to regulate competence development[21,22]. Conspicuously, Rap–Phr systems have not been thoroughly investigated with regards to biofilm formation[11]; to the best of our knowledge, only RapP has been previously shown to affect their formation[23,24].

Here, we aimed at revealing the influence of cultivation time (2 versus 5 days) and condition (planktonic versus biofilm) on selection toward particular *rap–phr* mutants or combined double mutations. To address this complexity, we used a barcoding approach that allowed us to follow the relative abundance of 79 strains, wild-type, single, and double *rap–phr* deletion mutants, within the populations subjected to various selective conditions.

## Results

**Multi-strain experimental conditions to determine fitness differences among *rap–phr* mutants of *B. subtilis*.** We were interested in systematically determining the impact that each Rap system (Fig. 1a) has on the population fitness of *B. subtilis*, particularly on sporulation within biofilms, and how absence or presence of different Rap phosphatases would affect the adaptability of *B. subtilis* to different growth conditions. We used *B. subtilis* DK1042 (hereafter WT), which is a transformable strain derived from the isolate NCIB 3610 harboring a *comI*[Q12I] mutation[25]. Strains NCIB 3610 and DK1042 have not undergone the domestication process that other commonly used *B. subtilis* strains have. This domestication can lead to loss of genetic functions and regulation changes, including modifications to Rap–Phr systems[26,27]. The WT has 11 *rap* genes encoded in its genome (*rapA–rapK*), additionally, it also possesses one more Rap–Phr system (*rapP–phrP*) encoded in its pBS32 plasmid[11,23]. We created single knockout mutant strains of all the *rap–phr* genes, and double knockout mutant strains that lack two *rap–phr*

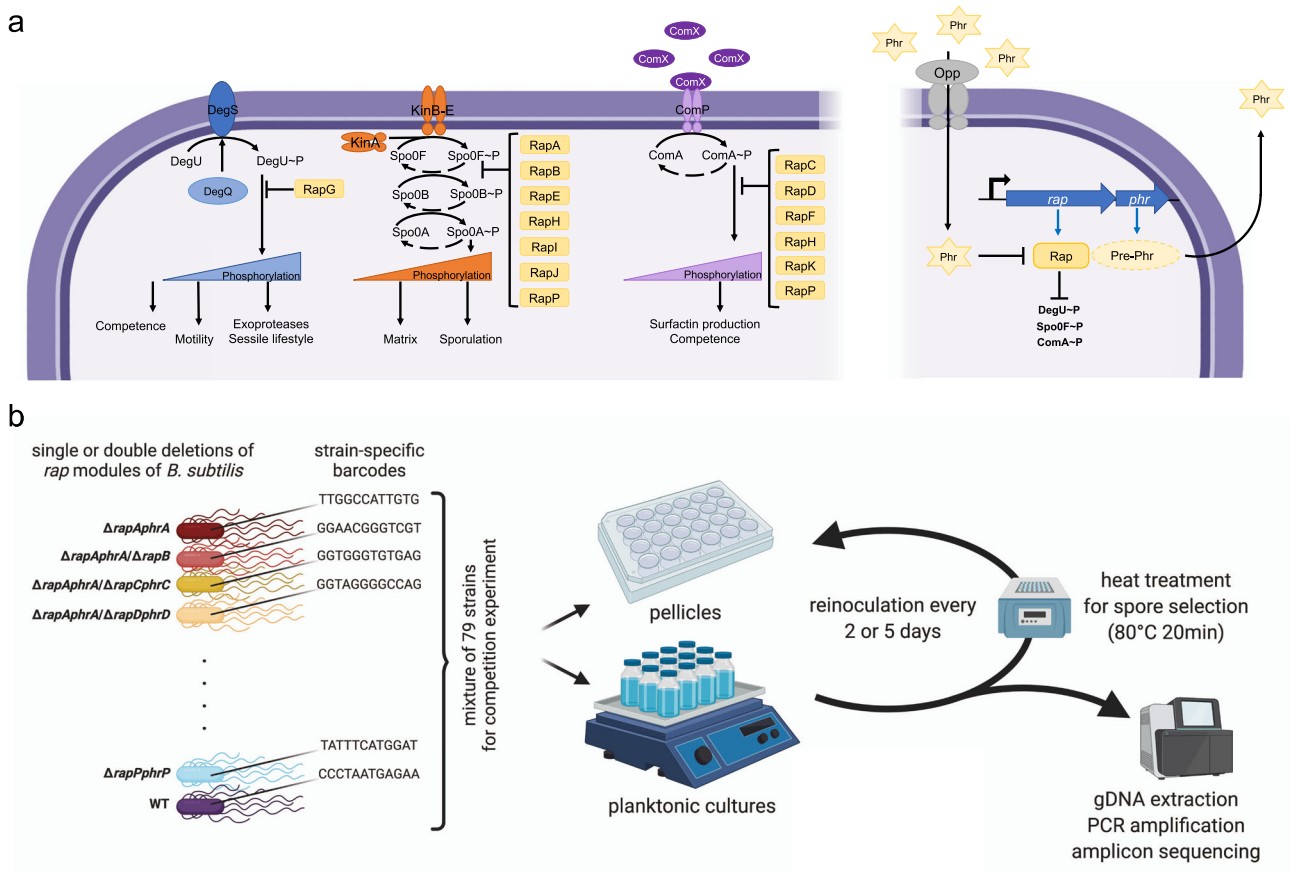

**Fig. 1 Overview of Rap–Phr systems, Rap-influenced global regulatory pathways, and the experimental setup.** The left side of panel (**a**) depicts the three main regulatory pathways (DegS–DegU, Spo0A phosphorelay, and ComP–ComA from left to right, respectively) affected by Rap proteins. The right side of panel (**a**) indicates the Rap–Phr pathway, where processed and secreted Phr acts as a quorum sensing signal, which after uptake inhibits the function of the corresponding Rap proteins. T-ended lines depict inhibition, black arrows indicate activation, blue arrows symbolize transcription and translation. Panel (**b**) represents the main steps of the competition experiment performed in this study; created with Biorender.com.

gene pairs in all possible combinations. In all cases both the *rap* and *phr* genes were deleted from the genome or plasmid. All created strains and the WT were further tagged with a pre-identified specific DNA barcode: a randomly generated 12 bp nucleotide sequence that was integrated into the *amyE* locus of each strain. We used an experimental competition approach to analyze how the different Rap–Phr system combinations would impact the adaptability of all mutant strains (and WT as control) to two different growth conditions: shaken liquid cultures, where cells would multiply in a planktonic state; or static liquid cultures, where cells would form a pellicle biofilm on the air-liquid interface (Fig. 1b). All studied strains (78 mutants + WT) were mixed together in equal ratios, and the mix was used to inoculate bottles (for planktonic cultures), or microplate wells (for pellicle cultures) with MSgg liquid medium. We introduced further variability to our studied conditions by using two different incubation times: 2 or 5 days, at 30 °C, for both culture conditions. Importantly, the secreted Phr peptides by any of the 79 competing strains are available for all the cells in these mixtures, indicating that the impact of *phr* deletions is less apparent in the competition setup. In contrast, the lack of Rap proteins specifically hinders the response of the cell to the specific Phr peptide and, more importantly, affects the activity of the target master regulator. We therefore notice that rather than studying the impact of quorum sensing, the impact of the Rap phosphatases on adaptability to different growth conditions are studied in this large competition experiment. After each incubation period, we collected spores

from the resulting cultures and used them to reinoculate bottles (for spores obtained from planktonic cultures) or microplate wells (for spores obtained from pellicle cultures) that were then incubated under the same conditions. The competition experiment was maintained for a total of nine reinoculation cycles for each culture condition. Full details of the competition experiment can be found in the "Methods." We purified total DNA from the cultures obtained after the first, third, fifth, seventh, and ninth culture cycle, and from the mix used to initiate the competition. The DNA was used to PCR amplify the *amyE* locus containing the strain-specific DNA barcodes. Using high-throughput sequencing, we were able to dissect the population dynamics of all the used strains throughout the competition experiment by analyzing their representation ratios within the competing population.

**Incubation time alters selection pressure on both planktonic and biofilm populations.** Experiments performed under all four studied growth conditions created distinct selection regimes, where different strains were favored during the competition (Fig. 2 and Supplementary Fig. 1). We expected that due to the specific culture reinoculation regime, cells that had formed mature spores within the available time (2 versus 5 days) would be transferred to the next culture iteration, and that those that multiplied for as long as possible prior to sporulation (thus delaying the spore maturation process, but still creating spores

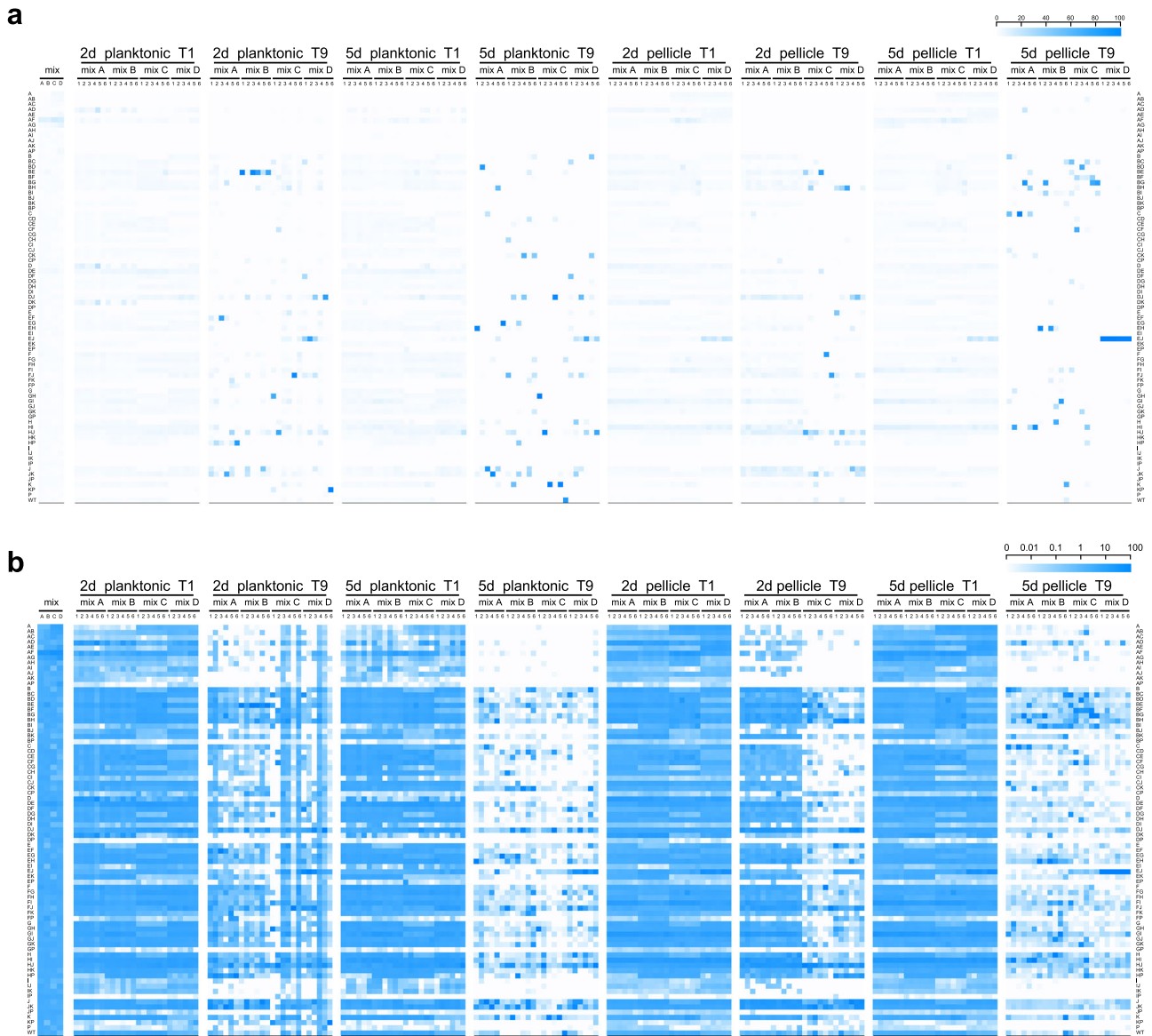

**Fig. 2 Heat map representation of the population dynamics of *B. subtilis rap–phr* mutants in competition.** Color intensities of the boxes represent the population percentage of strains. Text columns at far-left and far-right indicate which *rap–phr* genes have been deleted (A indicates a Δ*rapA* mutant, AB indicates a Δ*rapA*Δ*rapB* mutant, and so on), WT indicates *B. subtilis* DK1042. Text rows on top indicate type of culture (planktonic or pellicle), incubation period (2d = 2 days, 5d = 5 days), transfer number of represented population (t1–t9), and mix and replicate number. Competition populations were started from four biologically independent population mixes (A–D), with six technical replicates per mix. The first four box columns indicate the population representation of tested strains in the competition starter mixes. Both panels show the same data, although using different scales (shown in the top-right corner of each panel): **a** linear increment of percentage, **b** logarithmic increment of percentage.

before the applied bottleneck), would face a selection advantage due to higher spore numbers. We expected that this would reinforce the competitive selection of each culture condition, and thus amplify the effects of small advantages that the absence or presence of particular Rap–Phr systems would confer in regards to spore formation. Indeed, we observed that after the first culture transfer no strain had increased their total population representation beyond 10% in most experimental replicates (from initial average population representations of 1.27%). In contrast, at the ninth culture transfer the majority of the experimental replicates showed at least one strain that represented more than 25% of the total population (n = 11 for 2-day planktonic, n = 24 for the 5-day planktonic, n = 17 for the 2-day pellicle, and n = 24 for the 5-day pellicle conditions). Importantly, no correlation was detected in the variation of each strain between the initial

inoculation mixtures and after the ninth transfer in each selective condition (Spearman's rank correlation, adjusted *P* values of 0.38896, 0.8108, 0.11516, and 0.09132 for 2-day planktonic, 5-day planktonic, 2-day pellicle, and 5-day pellicle, respectively; Supplementary Fig. 2). In addition, to test if mixes had an effect on the experiment, a principal component analysis (PCA) was performed on data aggregated by replicate means showing that data points did not cluster according to mix (Supplementary Fig. 2E). This was further supported by PERMANOVA showing high explanatory power for genotype ($r^2 = 0.393$, $P = 0.001$), whilst mix, in contrast, had none ($r^2 = 0.00$, $P = 1.00$). Overall, this suggests that the major contributor to the observed variation is the genotype rather than the mix. One particular mutant (Δ*rapE*Δ*rapJ*) was strongly selected after nine transfers; however, only in mixture D, which suggests possible preexisting mutation in the

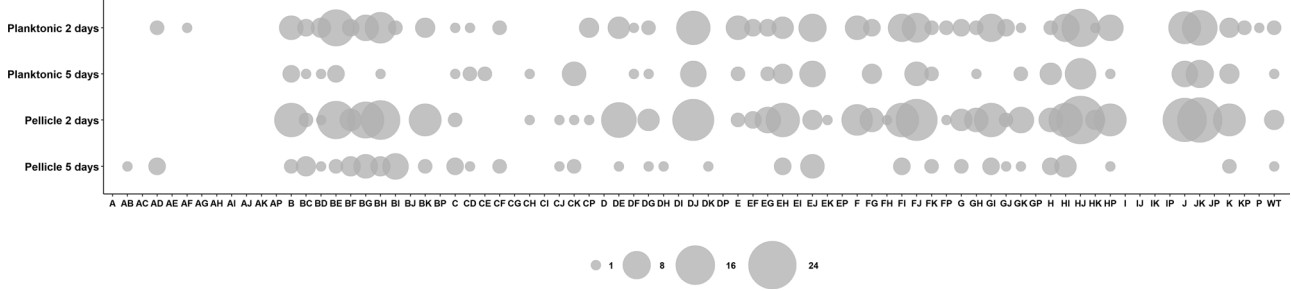

**Fig. 3 Bubble graph representation of mutant frequencies in the evolved populations.** Bubble sizes represent the number of populations (from $n = 24$) in which the given *rap–phr* mutant has a frequency higher than the putative input (i.e., 1.27%).

overnight grown culture benefiting this particular strain at the time of inoculation. However, this mutant was not increased similarly in other mixtures. Interestingly, strains that slightly increased their relative abundance in the population after the first culture transfer did not necessarily maintain this trend throughout the competition experiment, in both planktonic- and pellicle-forming conditions, e.g., Δ*rapA*Δ*rapD* or Δ*rapA*Δ*rapF* mutants (Fig. 2). This is probably caused by the evolutionary adaptation to the experimental conditions of other strains in the same population, and genetic drift, which may confer them a selective advantage independent from the Rap–Phr systems. Alternatively, initial population mix conditions, where all strains are present in similar ratios, may confer slight advantages to specific strains that are overcome by others during the competition. Importantly, the first growth round selects for fitness under planktonic or biofilm conditions, while in subsequent cycles, abundance highly depends on the sporulation in the previous cycle. In both planktonic- and pellicle-forming conditions incubation time was a critical selective parameter: populations that were incubated for 2 days showed greater variation in their final population composition than populations incubated for 5 days (Fig. 3 and Supplementary Fig. 3). The number of replicates, in which a certain mutant was able to maintain or exceed its original input percentage, i.e., was able to persist, was higher in the populations incubated for 2 days (see the size of the bubbles in Fig. 3). Thus, 5 days incubated populations were under a stronger selection pressure, where only a limited number of *rap–phr* mutant or double mutant strains were able to increase in relative abundance (Supplementary Fig. 3). Spore formation in *B. subtilis* begins with the detection of starvation conditions, however, this is not an homogeneous process in a population: cells use a bet-hedging strategy to avoid sporulation synchronization[28], furthermore, *B. subtilis* strains that lack Spo0F-specific Rap–Phr systems show temporal differences in sporulation initiation of several hours compared to strains that have those Rap phosphatases[17]. The 2-day incubation period seemed to be insufficient to trigger wide-spread sporulation, and thus only spores from mutants able to sporulate earlier were transferred into the next culture cycle. In contrast, the 5-day incubation seemed to allow specific strains to sporulate efficiently and thus be overrepresented at the start of each sequential culture cycle.

**Competition fitness does not depend on growth or biofilm formation exclusively**. Our experimental methodology and population dynamics analysis allows examination of fitness impact of each Rap–Phr deletion or combined deletion on sporulation under planktonic and biofilm conditions. Since population size determines the maximum number of possible spores, we also examined how the *rap–phr* mutations affect growth after 16 h of incubation (Fig. 4). We observed drastically different effects on growth depending on the mutated *rap* genes.

Interestingly, some *rap* mutations had a consistently negative effect on growth, but this effect was rescued in specific double *rap* mutants, e.g., a Δ*rapI* mutation had a strong negative effect on growth by itself or combined with any other *rap* mutation except Δ*rapG*, since a Δ*rapG*Δ*rapI* double mutant was able to grow almost as efficiently as WT (Fig. 4). On the other hand, certain single *rap* mutations that showed a mild effect or no effect on growth, such as Δ*rapA* and Δ*rapK*, resulted in a drastic growth defect when combined in a double mutant strain. Likewise, we examined the effect of the *rap* mutations during pellicle formation after 2 and 5 days of incubation (Supplementary Figs. 4 and 5). Again, drastic differences could be observed in the impact of single and double *rap* mutations upon the capacity of *B. subtilis* to form pellicles. Nevertheless, there was no correlation between growth and pellicle formation (compare Fig. 5 and Supplementary Figs. 4 and 5). Some strains that showed poor 16-h growth, such as Δ*rapB*Δ*rapI*, were able to form stronger pellicles compared with strains that grew more efficiently, such as Δ*rapB*Δ*rapH* (Fig. 5 and Supplementary Figs. 4 and 5). Furthermore, mutations in diverse Rap phosphatases that target the same regulator can have drastically diverse effects on *B. subtilis* competitiveness in our experimental conditions, both individually and epistatically: RapA and RapB regulate Spo0F, however, Δ*rapA* or Δ*rapA*Δ*rapB* strains became nearly extinct in all tested conditions, while a Δ*rapB* strain increased in frequency, especially when combined with mutations in other Rap proteins that also target Spo0F, such as Δ*rapB*Δ*rapE* and Δ*rapB*Δ*rapH* mutants. Efficient growth of a given mutant strain does not directly correlate with higher fitness in our competition experiments, e.g., a Δ*rapA*Δ*rapE* mutant strain shows better growth compared with a Δ*rapF*Δ*rapJ* strain, however, the later shows better competitiveness and increased population representation ratio under all studied conditions already after the first culture transfer. Importantly, the population obtained from the first growth cycle does not depend on viable spores yet, while all the subsequent cultures do. In accordance, all mutant strains that showed a drastic decrease in population representation after the first culture cycle also showed poor growth (Fig. 5). Thus, the regulatory and fitness impact of any individual Rap phosphatase cannot be understood solely by knowing its target transcriptional regulator; rather, the whole set of Rap–Phr proteins must be considered in order to explain the regulation of population heterogeneity in *B. subtilis*.

**Selection is defined both by deletion of selected *rap–phr* modules and acquired mutations**. The multi-strain competition experiment revealed no clear single winner strains for any of the four different growth conditions at the end of the competition experiment (Fig. 2). Furthermore, strains that increased their population representation after the first transfer did not in all cases maintain this trend throughout the experimental

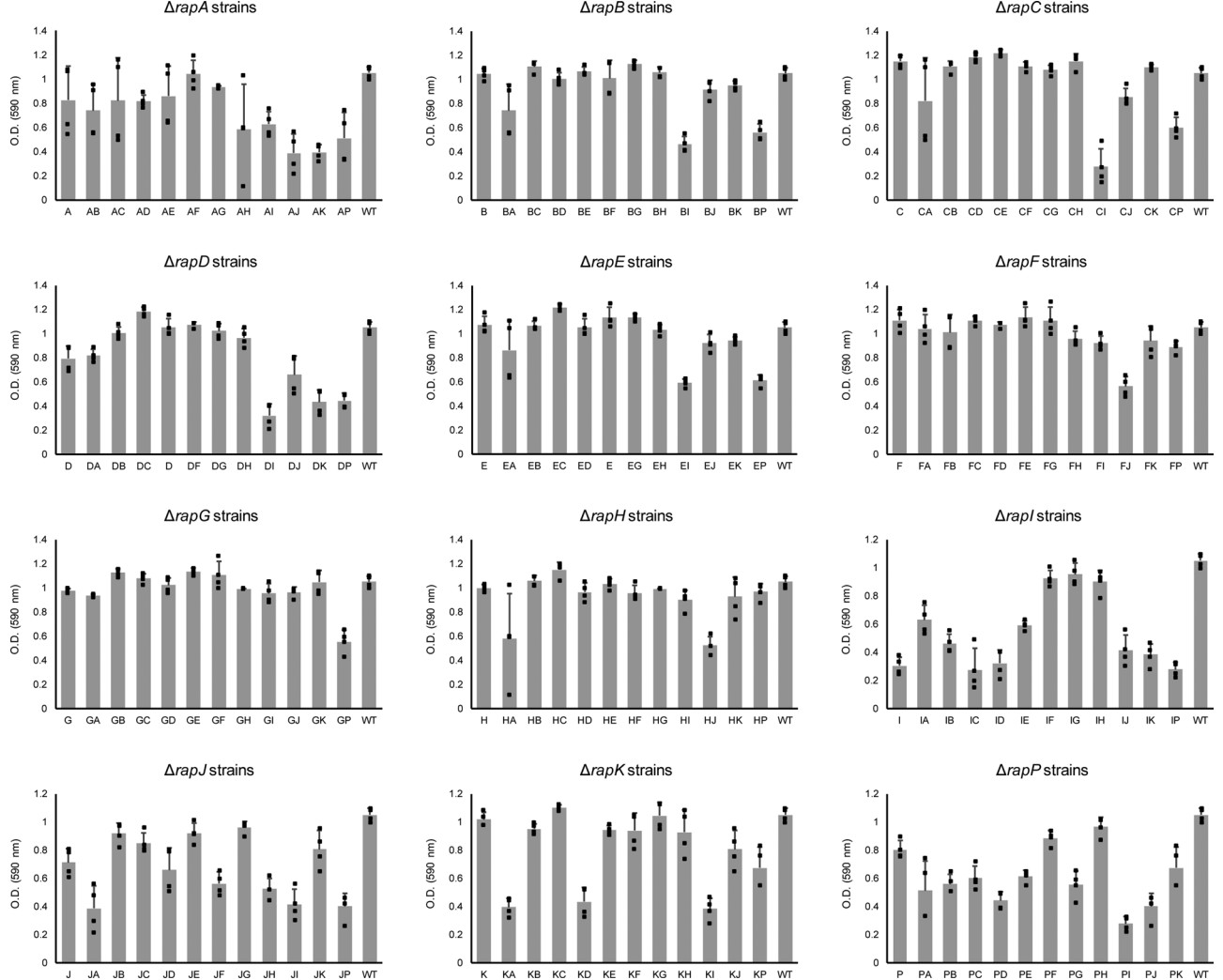

**Fig. 4 Growth of *B. subtilis* DK1042 and *rap–phr* mutants (as OD$_{590}$ increment after 16 h).** X-axis indicates which *rap–phr* genes have been deleted (A indicates a Δ*rapA* mutant, AB indicates a Δ*rapA*–Δ*rapB* mutant, and so on), WT indicates *B. subtilis* DK1042. Bars represent the average of four independent replicates. Error bars represent standard deviation.

competition (Fig. 2). Based on these observations, we speculated that the population dynamics were not solely driven by the *rap–phr* deletions. To scrutinize the genomic changes, ten evolved isolates representing all four competition conditions were randomly selected for complete genome resequencing at the end of the selection experiment. Resequencing data showed that eight out of ten isolated evolved strains had gained mutations in one or more genes involved in the regulatory network controlling population heterogeneity (Supplementary Data 1) besides additional mutations. Non-synonymous mutations were detected in *degU* (V278G in 5-day planktonic culture) and the biofilm matrix regulator *sinR* (L74S in 2-day pellicle culture) genes, the *comA* gene contained a frame shift mutation in one of the isolates from 2-day pellicle culture. In addition, mutations were identified in genes related to sporulation and germination, including *gerKA* (3 isolates), *gerKC* (2 isolates), *gerAB* (1 isolate), and also in the noncoding region upstream of *yhaX*, a gene that codes for a SigE-dependent sporulation protein. Finally, the pBS32 plasmid, therefore also the *rapP–phrP* genes, was lost in one of the 5 days cultured pellicle isolates that contained *rapB*, *rapD–phrD* deletions. None of the deleted *rap–phr* modules were restored in any of the isolates (e.g., driven by natural competence). Additional mutations were also detected unrelated to sporulation or with unknown function (Supplementary Data 1).

Importantly, nine out of the ten isolated evolved strains spontaneously released phage particles that showed lytic activity toward the ancestral strain (Supplementary Fig. 6A, B)—a phenomenon, recently described to be associated with amplification of cryptic phi3Ts and its recombination with indigenous SPβ (into phi3Ts-SPβ hybrids)[29]. Indeed, testing the culture supernatants of these evolved isolates against the indicator strain showed the presence of plaque formation (Supplementary Fig. 6A). In addition, the culture supernatant contained phage particles reminiscent of phi3Ts or its hybrids (Supplementary Fig. 6B), and testing phi3Ts-specific gene PCR verified the presence of the *rapX* gene in these nine isolates. We recently demonstrated that the cryptic phage, phi3Ts, a very close relative of phi3T (KY030782.1; 99.98% sequence identity), increases in copy number in various 3610 or domesticated 168 derivatives of *B. subtilis* when a sporulation bottleneck is used for selection. However, the direct role of the Rap-homolog in phi3T on sporulation or production of high-quality spores that germinate faster is unknown[29], as is the target of RapX. However, we hypothesized that in addition to such a phi3T-driven genetic adaptation that easily spreads around in the whole population, selection might be still driven by certain *rap–phr* deletions.

To test the hypothesis of the population dynamics being affected both by the original *rap–phr* mutations, as well as by the

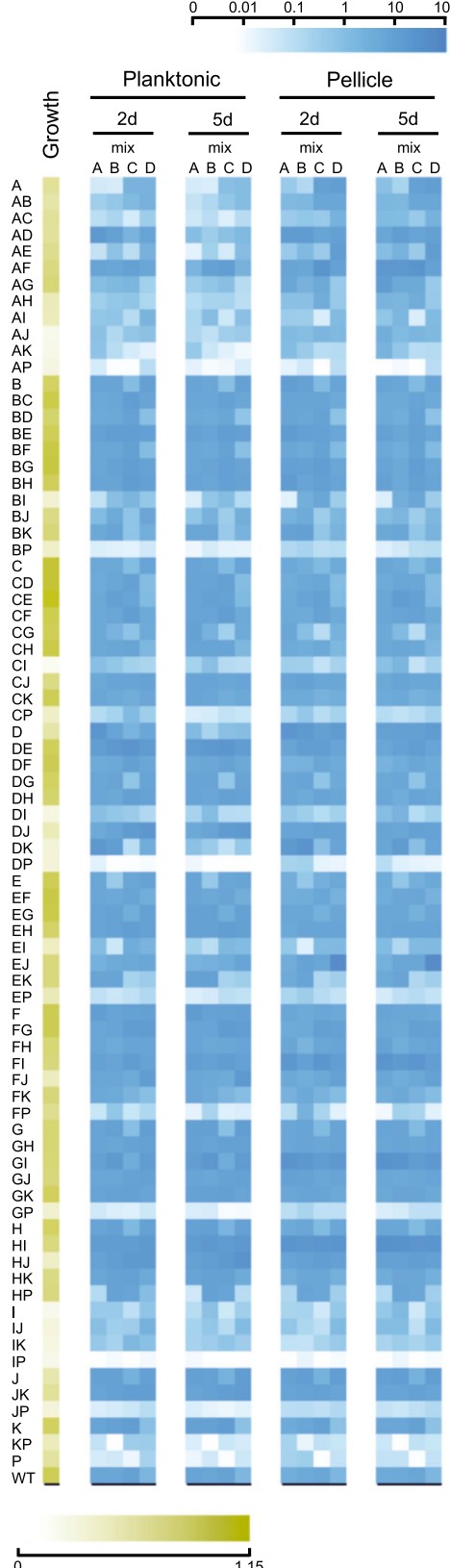

**Fig. 5 Heat map comparing growth of *B. subtilis rap–phr* mutants (16 h) to their population percentage after the first culture cycle (t1).** Yellow boxes represent 16-h growth on MSgg medium as increase in O.D.$_{590}$. The yellow intensity scale bar (bottom) indicates O.D.$_{590}$ increment. Blue boxes represent the population percentage of all tested strains. The blue intensity scale bar (top) indicates increment of percentage. Each blue box represents the average percentage of six population replicates per starter mix. Text rows on top indicate type of culture (planktonic or pellicle), incubation period (2d = 2 days, 5d = 5 days), and starter mix (A, B, C or D). Text columns at far-left indicate which *rap–phr* genes have been deleted (A indicates a Δ*rapA* mutant, ab indicates a Δ*rapAB* mutant, and so on), WT indicates *B. subtilis* DK1042.

strains were chosen based on their persistence after the ninth transfer under 2-day pellicle conditions, i.e., they showed higher final population representation than the input level (>1.27%, Fig. 2). Furthermore, the evolved strains isolated at the end of the competition were individually competed against their cognate ancestor in order to test whether the extra mutations arisen during the experiment affected fitness and sporulation. Importantly, we note that these control experiments do not represent the competition experiment of the 79 strains, however, these competitions provided novel knowledge about the individual impact of the *rap–phr* mutations as well as the extra mutations identified in the evolved strains on fitness and sporulation. The competitions were performed in a 1:1 starting ratio for 48 h to obtain pellicle biofilms. Afterwards, total cell suspensions as well as heat-treated cell suspensions (to obtain spores) were plated for CFU counting, and the relative fitness (of all cells) and sporulation fitness (SF) of ancestor versus WT, and of evolved versus ancestor was calculated (see "Methods" for calculations).

The Δ*rapB*, Δ*rapB*Δ*rapD*, Δ*rapB*Δ*rapE*, Δ*rapB*Δ*rapH*, Δ*rapC*, Δ*rapE*Δ*rapF*, Δ*rapE*Δ*rapH*, Δ*rapH*Δ*rapI*, and Δ*rapH*Δ*rapP* ancestors showed significantly increased fitness and/or SF compared to the WT (Fig. 6a, b). The Δ*rapB*Δ*rapJ* ancestor was not affected in fitness nor SF compared to the WT, while the Δ*rapD*Δ*rapJ*, Δ*rapF*Δ*rapJ*, Δ*rapH*Δ*rapJ*, Δ*rapJ*, and Δ*rapJ*Δ*rapK* ancestors showed reduced pellicle fitness or SF compared to the WT. Of the evolved isolates, Δ*rapB*Δ*rapD*, Δ*rapC*, Δ*rapE*Δ*rapF*, Δ*rapH*Δ*rapI*, and Δ*rapJ*Δ*rapK* demonstrated significantly enhanced spore fitness compared to their ancestor (Fig. 6c, d). The evolved Δ*rapB*Δ*rapE* exhibited enhanced pellicle fitness but reduced spore fitness, while the evolved Δ*rapE*Δ*rapH* and Δ*rapH*Δ*rapP* were not affected in either, compared to their ancestor.

The results of these competition experiments reveal that both the original *rap–phr* deletion mutations as well as the extra mutations arisen during the nine transfers can have an impact on the fitness and sporulation of the strains and thereby also on the observed dynamics of the 79-strain population competition under sporulation selection regime.

## Discussion

In this work, we have examined how variability in the number and function of Rap–Phr pairs allows *B. subtilis* to adapt to certain selection pressures. Our experimental competition approach, paired with high-throughput sequencing, allowed us to analyze the population dynamics during a multi-strain competition and assess the impact of each Rap–Phr system. Furthermore, the use of single and double *rap–phr* mutants revealed epistatic effects that the presence or absence of specific Rap–Phr systems may have upon population differentiation of *B. subtilis*, including biofilm development and spore formation. Recent investigations indicate that Rap–Phr systems are readily transferred among *B. subtilis* strains, possibly helped by the natural development of

adaptation of some strains to the growth conditions (either by acquiring additional mutations, or by phi3T-driven gene expression), two different competition experiments were performed. To test the impact of the original *rap–phr* mutations on fitness and sporulation, some of the ancestor Δ*rap* mutant strains were competed independently against the WT (Fig. 6). The competed

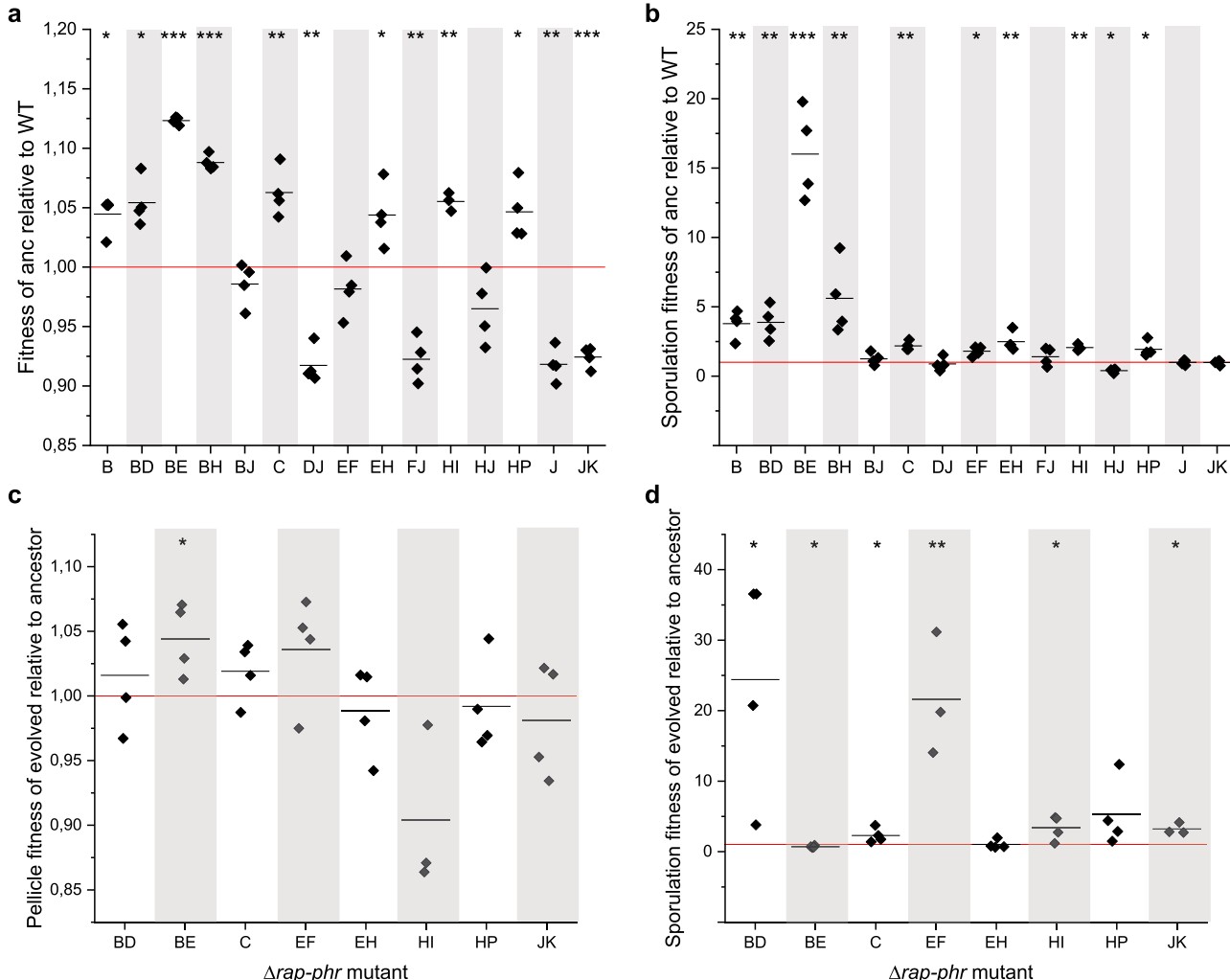

**Fig. 6 The impact of *rap-phr* deletions and additional mutations arisen during the experiment on fitness and sporulation.** To test the impact of the *rap–phr* deletions on fitness and sporulation, mutants showing persistence in several replicates after transfer 9 (>1.27% abundance) were competed against the WT in MSgg medium to allow pellicle development. Similarly, to test the effect of the extra mutations arisen during the experiment on fitness and sporulation, the isolated evolved strains were competed against their cognate ancestor in MSgg medium. After 2 days, the developed pellicles were harvested, sonicated, and exposed to heat treatment in order to kill vegetative cells. Total cell suspensions as well as heat-treated cell suspensions were plated to quantify total cell number and number of spores. **a** Relative fitness of the ancestor *rap–phr* mutant strain compared to WT. **b** Sporulation fitness of the ancestor *rap–phr* mutant strain compared to WT. **c** Relative fitness of evolved *rap–phr* mutant strain compared with its cognate ancestor. **d** Sporulation fitness of the evolved strains compared to its cognate ancestor. Data are represented with technical replicates ($n = 4$) as squares and the mean of the replicates are indicated as a black line ($n = 3$–4). The red horizontal line denotes the WT (**a**, **b**) or the cognate *rap–phr* mutant ancestor (**c**, **d**). For statistical analysis, the relative (sporulation) fitness was log-transformed and a one-sample *t*-test was performed to test whether the mean was significantly different from 0. *$P < 0.05$, **$P < 0.005$, ***$P < 0.001$.

competence by this bacterium and by the ability of some Rap–Phr systems to promote their own genetic mobility[15,16]. Our results indicate that variability in Rap–Phr systems among *B. subtilis* strains fine-tunes their ability to compete in diverse environments, and that Rap–Phr systems are particularly important if adaptation depends on differentiation of particular cell types, such as spores. Over evolutionary time, and paired with the genetic plasticity of this bacterium, this could explain the genomic diversity of Rap–Phr systems and the ecological success of this widely spread bacterium[15,16].

Our initial hypothesis was that the strains initially showing increased population representation in the first culture and subsequent sporulation would be the ones adapting most efficiently to the growth-sporulation cycles applied during our competition, and thereby be the ones winning (having a large increase in their population proportion) at the end of the experiment.

Interestingly, no single strain was detected to be a transcendent winner for all of the four different conditions at the end of the competition experiment. Instead, various *rap–phr* mutant strains show an increase in population abundance under specific competition conditions, e.g., multiple population replicates grown under planktonic culture conditions (for 5 days) show strains *ΔrapDΔrapJ* and *ΔrapHΔrapJ* to increase their population abundance, but this is not the case for the populations grown under pellicle-forming conditions for 5 days. On the other hand, mutants in specific Rap–Phr systems do show consistently decreased population abundance across all tested conditions, e.g., most double *rap–phr* mutants with a *ΔrapK* deletion. Our competition approach allows us to compare the effect of particular *rap–phr* mutations across different conditions, and thus analyze their regulatory function: RapK has been previously suggested to inhibit ComA activity[14,19], here we show that various *ΔrapK*

strains when harboring other *rap* deletions (e.g., Δ*rapA*, Δ*rapD*, Δ*rapE*, Δ*rapF*, Δ*rapH*) suffer a loss of competition fitness when selecting for spores. We hypothesize that the Δ*rapK* mutation leads to an overexpression of ComA-controlled genes related to production of antimicrobials[30], exoenzymes[31], and competence development[32], this, in turn, may direct metabolic resources away from spore production and maturation, thus resulting in the observed loss of competition fitness, but only in combination with other Δ*rap* mutation. Another interesting behavior was that *rap–phr* mutants that increased their population representation after the first transfer did not necessarily maintain their abundance throughout the multi-strain competition experiment (Fig. 2). Under 2-day pellicle conditions, strains Δ*rapA*Δ*rapD*, Δ*rapB*Δ*rapE*, Δ*rapD*, Δ*rapD*Δ*rapK*, Δ*rapF*Δ*rapI*, Δ*rapG*Δ*rapI*, and Δ*rapH*Δ*rapI* showed an initial increase in abundance (i.e., during the first transfer). However, of those only the Δ*rapB*Δ*rapE*, Δ*rapF*Δ*rapI*, Δ*rapG*Δ*rapI*, and Δ*rapH*Δ*rapI* strains kept this trend throughout the experiment and showed increased population representation in most replicates after transfer 9. On the other hand, Δ*rapD*Δ*rapJ*, Δ*rapF*Δ*rapJ*, Δ*rapH*Δ*rapJ*, Δ*rapJ*, and Δ*rapJ*Δ*rapK* mutants did not show increased abundance after transfer 1, but were increased after transfer 9. This highlights the versatility that the regulatory effect of Rap–Phr systems confers to *B. subtilis'* ability to generate a phenotypically diverse population. Certain *rap–phr* mutant strains display an initially increased fitness when competing under conditions where all strains are present in the same ratio (starting conditions of the competition), while other strains show an increase in abundance over time. Their particular *rap–phr* mutations allows them to capitalize on hypothetically small fitness advantages when producing mature spores. Furthermore, during the sequential competition transfers, evolutionary adaptation to the experimental conditions of other strains in the population may take place, resulting in those strains gaining an increased SF, and possibly taking over the strains initially showing an increase in population abundance. In support of this, resequencing of ten clones isolated from random populations at the end of the competition experiment revealed that eight of the ten strains had gained mutations in different genes, including genes involved in the population-heterogeneity regulatory network, such as sporulation. This indicates that the strains can quickly acquire additional mutations that help them further adapt to the growth-sporulation cycles applied during the competition experiment.

Competition experiments quantifying fitness and spore fitness were employed to compare *rap–phr* mutants with WT and the evolved *rap–phr* mutants with their respective ancestors. For example, the Δ*rapB*Δ*rapE* ancestor showed enhanced fitness and SF compared to the WT after 2-day pellicle formation, while the evolved strain isolated from a 2-day pellicle culture at the end of the competition experiment (transfer 9) showed increased pellicle but reduced spore fitness compared to the ancestor. This evolved Δ*rapB*Δ*rapE* strain had gained mutations in two genes, *gerKC* and *yqcG* encoding a membrane receptor involved in spore germination[33] and a toxin[34], respectively. The Δ*rapB*Δ*rapE* mutant that maintained the increased abundance in the 2-day pellicle conditions throughout the nine transfers is therefore presumably a result of both deletion in Δ*rapB* and Δ*rapE*, as well as the mutations arising during the experiment. Another *rap–phr* mutants with positive fitness or spore fitness compared to the ancestor WT also acquired mutations that may have increased their fitness during the multi-strain competition experiment, e.g., mutation in *degS* in the Δ*rapH*Δ*rapI* background. The observation that the majority of the isolated evolved strains show enhanced SF compared to the cognate ancestor indicates that the mutations arising during the experiment and the acquired awakening of the phi3T phage have a dramatic impact on the SF,

thereby allowing the strains to adapt to the growth-sporulation cycles applied in the competition experiment. This, as well as the observation that most of the *rap* mutations have an effect on fitness and sporulation, indicates that the population dynamics during the experiment are a result of both the original *rap–phr* mutations present in the 78 mutant strains, as well as the additional mutations and phi3T arising during the experiment.

The differences in the variance of surviving *rap–phr* strains between 2- and 5-day culture conditions, and between planktonic- and pellicle-forming conditions points to dissimilar selection pressures generated by these growth conditions that further reveal the role that Rap–Phr systems play in environmental adaptability, focusing on the ability of cells to generate mature spores, and to form pellicles. Additionally, genetic drift and negative frequency-dependent selection may occur under the used conditions.

Our results indicate that variability in Rap–Phr systems among *B. subtilis* strains strongly impacts their ability to compete in diverse environments, and that these systems are particularly important if adaptation depends on differentiation of particular cell types such as spores. We propose that the diversity of Rap–Phr regulatory systems allows *B. subtilis* to fine-tune its genetic regulatory network in order to quickly adapt to new ecological niches, possibly accompanied by the acquisition of additional mutations that provide the bacterium with increased fitness. Furthermore, genetic exchange of elements of a family of regulatory proteins could be a general mechanism for genetically related bacteria to more efficiently and quickly adapt to new environments.

## Methods

**Media and cultivation methods.** When fresh cultures were needed, strains were pregrown overnight in Lysogeny broth (LB) medium (LB-Lennox, Carl Roth; $10\,g\,L^{-1}$ tryptone, $5\,g\,L^{-1}$ yeast extract, and $5\,g\,L^{-1}$ NaCl) at 37 °C and shaken at 225 rpm. LB medium was used for all *B. subtilis* and *Escherichia coli* transformations. MSgg medium (5 mM potassium phosphates buffer (pH 7), 100 mM MOPS, 2 mM MgCl$_2$, 700 μM CaCl$_2$, 100 μM MnCl$_2$, 50 μM FeCl$_3$, 1 μM ZnCl$_2$, 2 μM thiamine, 0.5% glycerol, 0.5% glutamate, 50 μM L-tryptophan, and 50 μM L-phenylalanine, adapted from[35]) was used for the competition experiment and to examine growth kinetics and pellicle formation. GCHE medium (1% glucose, 0.2% glutamate, 100 mM potassium phosphate buffer (pH: 7), 3 mM trisodium citrate, 3 mM MgSO$_4$, 22 mg L$^{-1}$ ferric ammonium citrate, 50 mg L$^{-1}$ L-tryptophan, and 0.1% casein hydrolysate) was used to induce natural competence in *B. subtilis*[36]. Gallegos Rich medium was used to grow *Lactococcus lactis* MG1363, in order to purify pMH66: 21 g L$^{-1}$ tryptone, 5 g L$^{-1}$ yeast extract, 8.3 g L$^{-1}$ NaCl, 3 g L$^{-1}$ soya peptone, 2.6 g L$^{-1}$ glucose, and 2.5 g L$^{-1}$ MgSO$_4$·7H$_2$O[37]. Media were supplemented with Bacto agar 1.5 % when media were needed for preparing plates. Antibiotics were used at the following final concentrations: kanamycin, 10 μg mL$^{-1}$; chloramphenicol (chl), 5 μg mL$^{-1}$; erythromycin-lincomycin (MLS), 0.5 μg mL$^{-1}$ and 12.5 μg mL$^{-1}$, respectively; ampicillin, 100 μg mL$^{-1}$; spectinomycin, 100 μg mL$^{-1}$; tetracycline, 10 μg mL$^{-1}$.

**Strain and plasmid construction.** All strains used in this study are listed in Supplementary Data 2. To create the single and double *rap–phr* mutant strains, plasmids were first designed that allowed to create clean-deletion mutants of all *rap–phr* gene pairs. All plasmids used in this study are listed in Supplementary Data 1, and they were created using standard molecular biology techniques. Briefly, upstream and downstream DNA fragments of ~600 bp flanking the *rap–phr* genes to be mutated were PCR amplified from genomic DNA of *B. subtilis* NCIB 3610. Afterwards, these DNA fragments were sequentially cloned into plasmids pTB120, pTB233, or pTB234, all of which are pBluescript SK(+)-derived plasmids containing an antibiotic resistance cassette (kanamycin, spectinomycin, and MLS, respectively) flanked by Cre recombinase recognition sites *lox66* and *lox71*. The respective antibiotic resistance genes have been amplified using oligonucleotides indicated in Supplementary Data 3 using pBEST501 (kanamycin)[38], pWK-Sp (spectinomycin)[39], and pDR183 (MLS)[40] as templates. Thus, the obtained plasmids contain an antibiotic resistance cassette flanked by the upstream and downstream regions of a given *rap–phr* gene pair. All plasmids were created and maintained in *E. coli* MC1061.

*B. subtilis* mutants of a single *rap–phr* pair were created via transformation of DK1042 with the corresponding plasmid containing the flanking regions of the target *rap–phr* pair. Double *rap–phr* mutants were created by transforming clean-deletion mutants of single *rap–phr* pairs with genomic DNA of strains that had the desired *rap–phr* mutation still with the corresponding antibiotic resistance cassette.

All *B. subtilis* strains generated in this work were obtained via natural competence transformation[36]. Briefly, overnight cultures of the receiver strains grown in LB medium were diluted to a 1:50 ratio with GCHE medium, these cultures were incubated at 37 °C for 4 h with shaking at 225 rpm. After this incubation period, 5–10 μg of genomic or plasmid DNA was mixed with 500 μL of competent cells and further incubated for 2 h before plating on LB plates added with selection antibiotics. *B. subtilis* clean-deletion mutants of single *rap–phr* gene pairs were obtained by using the Cre recombinase expressed from plasmid pMH66 to eliminate their corresponding antibiotic resistance cassette, and subsequently curing pMH66 via thermal elimination of the plasmid[41]. Briefly, strains were transformed with 10 μg of pMH66, selecting transformants via incubation at 37 °C on LB plates added with tetracycline. Candidates were then screened for their capacity to grow at 37 °C on LB plates added with the antibiotic to which their parental strains (prior to transformation with pMH66) were resistant, those that were not able to grow were further incubated on LB plates at 43 °C for 18 h to induce the loss of pMH66. Candidates that were then unable to grow at 37 °C on LB plates added with tetracycline were considered to have lost pMH66.

In order to track each strain during the competition experiment, *B. subtilis* DK1042 and all single and double *rap–phr* mutants were marked with a randomly generated DNA 12 bp barcode in their *amyE* locus. To do this, plasmid pTB666 was created by cloning a chl resistance cassette (*cat*) into pTB16[42], substituting its original kanamycin resistance cassette. The *cat* cassette was amplified from pNW33n using primers oTB118 and oTB119 (see Supplementary Data 3). Primer oTB119 has a 12 nt-long random sequence after the binding site of the primer, the DNA barcode. Thus, pTB666 carries a barcoded *cat* which is flanked by the 5′- and 3′-end of the *B. subtilis amyE* gene. Eighty different clones of *E. coli* carrying pTB666 were isolated during creation of this plasmid. Each version of pTB666 from these clones was isolated and sequenced with oBC_rev in order to identify it. The various versions of pTB666 were used to transform *B. subtilis* DK1042 and all single and double *rap–phr* mutants using natural transformation as described above.

Successful construction of all used strains and plasmids was validated via PCR, sequencing, and restriction pattern analysis; and by the lack of amylase activity on LB plates added with 1% starch for the case of barcoded strains[43]. All PCR primers used in this study are listed in Supplementary Data 3. Primer pairs were used to amplify the indicated loci in order to confirm the proper mutation of the corresponding gene.

**Multi-strain experimental competition**. The experimental competition was done using the barcoded versions of *B. subtilis* DK1042 and the single and double *rap–phr* mutants. The experiment was initiated from four different starter mixes. Each mix was obtained by mixing overnight cultures of all the competing strains in similar ratios after adjusting their optical density (O.D.) at 600 nm to 1.0. Each starting mix was used to inoculate 100 ml bottles (0.5 ml of mix + 9.5 ml of MSgg medium) and 2 ml microplate wells (100 μl of mix + 1900 μl of MSgg medium). The experimental competition used four growth conditions: planktonic growth (10 ml culture in bottles shaken at 200 rpm) or pellicle development (static 2 ml in 24-well microplate), and incubation for 2 or 5 days. Twenty-four replicate populations were used for each growth condition (six replicates from each starting mix). All cultures were incubated at 30 °C throughout the experiment.

After each incubation period, spores obtained from each population replicate were used to inoculate the next iteration of the same population. For this, pellicles obtained from the microplate cultures were collected in Eppendorf tubes with 1 ml of MSgg medium and sonicated until the pellicles were completely dispersed; afterwards, 100 μl aliquots from the dispersed pellicles, or 500 μl aliquots from the planktonic culture bottles were incubated at 80 °C for 20 min in order to kill all vegetative cells. After the incubation period, the heat-treated aliquots were used to inoculate new 100 ml bottles or 2 ml microplate well using the same volumes as during the initiation of the experiment. This regime was followed during nine culture reinoculation cycles, resulting in >39 generations. Additionally, DNA was obtained from aliquots of the starter mixes and from aliquots of each population replicate obtained before the heat treatment during the first, third, fifth, seventh, and ninth culture reinoculations. The DNA extraction was performed with the GeneMatrix Bacterial and Yeast Genomic DNA Purification Kit (EURx Ltd., Poland) with the following modifications to the manufacturer's instructions: step 2, added 10 μl of lysozyme (10 mg ml⁻¹); step 3, extended the incubation time to 25 min; step 6, extended the incubation time to 45 min.

**Forty-eight-plex high-throughput barcode sequencing**. The *B. subtilis amyE* locus containing the barcodes was PCR amplified from the DNA samples obtained from the competition experiment using primers oBC1–oBC4 and oBC5–oBC16. These primers contain distinct 5-bp (for oBC1–4 primers) or 7-bp (for oBC5–16 primers) sequences that allowed us to identify up to 48 individual replicate populations per Illumina sequencing run. Data analysis was carried out using the R statistics environment[44]. PCR products, each represented by one R1-R2 Illumina sequence pair, were looked up for the presence of the 79 barcodes that differentiate between the 79 bacterial strains used in the study (see Supplementary Data 4 for the barcode sequences of each strain). We linked a PCR product to a given barcode if at least one of its paired-end reads displayed 100% sequence identity over the entire length of the barcode. PCR products that gave ambiguous results (i.e., hits

against multiple barcodes) were excluded from the study. Figures 1 and 3 were prepared with Genesis[45].

**Growth kinetics measurements**. We examined the ability of all barcoded strains to grow on MSgg medium in order to assess the impact of the *rap–phr* mutations. Overnight LB liquid cultures of all barcoded strains were adjusted to O.D.600 0.2. 10 μl of the O.D.-adjusted cultures was added to 190 μl of MSgg liquid medium in 200 μl microplate wells. These cultures were incubated at 30 °C for 17 h with shaking. Cell growth was measured as O.D.590 change every 15 min using a Tecan Infinite 200Pro microplate reader (Tecan Group Ltd., Switzerland).

**Pellicle formation**. We examined the ability of all barcoded strains to form pellicles on MSgg medium. Overnight LB liquid cultures of all barcoded strains were adjusted to O.D.600 0.1. 20 μl of the O.D.-adjusted cultures was added to 2 ml of MSgg liquid medium in 2 ml microplate wells. These cultures were incubated at 30 °C for 5 days. After 2 days of incubation and at the end of the incubation period the obtained pellicles were examined with an Axio Zoom V16 stereomicroscope (Carl Zeiss, Germany) equipped with a Zeiss CL 9000 LED light source, a PlanApo Z ×0.5 objective, and AxioCam MRm monochrome camera (Carl Zeiss, Germany).

**Resequencing of selected clones**. Ten populations from the end of the competition experiment were selected randomly (representing at least two replicates of each growth condition). Aliquots of the selected populations were used to inoculate LB plates that were incubated overnight at 30 °C in order to isolate clones from each population. Overnight LB liquid cultures of the isolated clones were used to extract genomic DNA using the GeneMatrix Bacterial and Yeast Genomic DNA Purification Kit (EURx Ltd., Poland). Paired-end libraries were prepared using the NEBNext® Ultra™ II DNA Library Prep Kit for Illumina. Paired-end fragment reads were generated on an Illumina NextSeq sequencer using TG NextSeq® 500/550 High Output Kit v2 (300 cycles). Primary data analysis (base-calling) was carried out with "bcl2fastq" software (v2.17.1.14, Illumina). All further analysis steps were done in CLC Genomics Workbench Tool 9.5.1. Reads were quality-trimmed using an error probability of 0.05 (Q13) as the threshold. In addition, the first ten bases of each read were removed. Reads that displayed ≥80% similarity to the reference over ≥80% of their read lengths were used in mapping. Nonspecific reads were randomly placed to one of their possible genomic locations. Quality-based SNP (single nucleotide polymorphism) and small In/Del variant calling was carried out requiring ≥8× read coverage with ≥25% variant frequency. Only variants supported by good quality bases ($Q ≥ 20$) were considered and only when they were supported by evidence from both DNA strands.

**Competition experiments**. For competition experiments in pellicles, overnight LB liquid cultures of ancestors, evolved strains, and WT labeled with mKATE were adjusted to same OD600. The WT strain was labeled with mKATE to facilitate counting of WT colonies on agar plates after the competition experiments. None-barcoded ancestors were competed against WT, whereas the evolved, barcoded strains, isolated after transfer 9, were mixed with their cognate (none-barcoded) ancestor strain in a 1:1 ratio. From this mix, 15 μL was added to 1.5 mL MSgg liquid medium in 2 mL microplate wells in a 24-well plate. The cultures were grown under static conditions at 30 °C for 48 h. After incubation, the pellicles were harvested and transferred into Eppendorf tubes containing 1 mL NaCl 0.9% (Carl Roth) and subjected to standard sonication protocol allowing proper disruption of the biofilm without damaging the cells. Afterwards, 200 μL of the obtained cell suspensions was incubated at 80 °C for 20 min, killing all vegetative cells leaving only spores left. CFU assays were performed for cell suspensions (total cells) and heat-treated aliquots (spores only). For the ancestor versus WT competition, obtained cell suspensions were plated on LB plates for CFU counting. For the evolved versus ancestor competition assays, cell suspensions were plated on two different types of plates: LB plates and LB + chl (5 μg ml⁻¹). LB medium allowed growth of both evolved and ancestors, while LB + chl only allows growth of evolved strains due to the presence of a *chl* resistance gene next to the barcode gene in the evolved strains. Thereby, CFU of evolved strains equals the CFU obtained from the LB + chl plates, while CFU of ancestors equals the CFU on the LB plates (evolved + ancestors) subtracted the CFU of LB + chl plates (evolved only).

Relative fitness was calculated for ancestor strains relative to the mKATE-labeled WT, and for evolved strains relative to their cognate ancestors. Relative fitness for strain A, $W_A$, in competition with strain B was calculated as follows:

$$W_A = \frac{\ln\left(\text{CFU}_{A_{48h}}\right) / \ln(\text{CFU}_{A\_start})}{\ln\left(\text{CFU}_{B_{48h}}\right) / \ln(\text{CFU}_{B\_start})} \tag{1}$$

In addition, SF of ancestor strains relative to WT and of evolved strains relative to ancestors was calculated. Relative SF for strain A ($\text{SF}_A$) in competition with strain B was calculated as follows:

$$\text{SF}_A = \frac{\text{CFU spores}_{A_{48h}} / \text{CFU cells} + \text{spores}_{A\_start}}{\text{CFU spores}_{B_{48h}} / \text{lCFU cells} + \text{spores}_{B\_start}} \tag{2}$$

**Statistics and reproducibility**. Statistical analysis was performed using OriginPro 2018b (OriginLab, USA). The relative fitness values calculated from the competition experiments were log-transformed, and statistical differences from $\log10(W)$ = 0 (corresponding to statistical differences of the relative fitness values from $W = 1$) were calculated using one-sample $t$-test with test mean = 0 and a significance level of 0.05. Prior to using $t$ tests, normality was tested by Shapiro–Wilks test. PCA on aggregated and scaled data was used to visualize the multivariate pattern of each genotype stratified by mix. Analogously, PERMANOVA, using Euclidian distances, was used to test the statistical effect of both these variables. No statistical tests were used to determine sample sizes. Such was chosen based on experience and knowledge on expected biological variability (which cannot be controlled by the experimenter) and accuracy of the experimental methods used. No data were excluded from the analyses. All experiments contained biological replicates with specific information on replicates in the "Methods," and the vast majority of the experiments were performed multiple times.

**Plaque assay and phage purification**. Supernatants, obtained from overnight cultures of the evolved strains, were spotted on soft-agar (0.3%) lawn of the ancestor strain with knockout of biofilm genes (NCBI 3610 $\Delta eps\Delta tasA$). Biofilm mutant was used as a lawn to improve the visibility of plaques. Phage particles were purified like described previously[46]. Specifically, culture supernatants were collected, mixed at a 1:4 ratio with PEG-8000 solution (PEG-8000 20%, 2 M NaCl), incubated on ice for at least 90 min and precipitated by centrifugation (20 min, 7600 rpm). The pellet was resuspended in 10% of the original supernatant volume in TBS solution (50 mM Tris-HCl, 150 mM NaCl, pH 7), incubated on ice for 90 min and centrifuged (20 min, 7600 rpm). Supernatant was carefully transferred to clean Eppendorf tubes and phages were visualized using electron microscopy.

**Transmission electron microscopy**. Before use, 400 mesh nickel grids with a 3–4 nm-thick carbon film, CF400-Ni-UL EMS Diasum, were hydrophilized by 30 s of electric glow discharging. Next, 5 µl of purified phage solutions were applied onto the grids and allowed to adsorb for 1 min. The grids were rinsed three times on droplets of milliQ water and subjected to staining with 2% uranyl acetate. Specifically, with a help of EM grid-grade tweezers, the grids were placed sequentially on droplets of 2% uranyl acetate solution for 10, 2, and 20 s. Excess uranyl acetate was wicked away using filter paper and the grids were allowed to dry overnight and stored in a desiccator until analysis. Transmission electron microscopy was performed utilizing a FEI Tecnai T12 Biotwin TEM operating at 120 kV located at the Center for Electron Nanoscopy at the Technical University of Denmark, and images were acquired using a Bottom-mounted CCD, Gatan Orius SC1000WC.

**Reporting summary**. Further information on research design is available in the Nature Research Reporting Summary linked to this article.

## Data availability
Raw sequencing data have been deposited to NCBI Sequence Read Archive (SRA) database. Bioproject accession number: PRJNA626081. Data used to create Figs. 2–6 are provided as Supplementary Data 5.

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

## Acknowledgements

We thank Mikael Lenz Strube for his suggestions on statistics and PCA. R.G.-M. was supported by Consejo Nacional de Ciencia y Tecnología (CONACyT), German Academic Exchange Service (DAAD). Á.T.K. was supported for this project by a Marie Skłodowska Curie Career Integration Grant (PheHetBacBiofilm), DFG Graduate School Jena School of Microbial Communication (JSMC), and a DTU Bioengineering Start-up fund. This project has received funding from the European Union's Horizon 2020 research and innovation program under the Marie Skłodowska-Curie Grant Agreement No. 713683 (H.C. Ørsted COFUND to A.D.).

## Author contributions

Á.T.K. and R.G.-M. conceived the project; R.G.-M., M.N.C., T.B., S.K., B.P., and P.J.K. performed experiments; B.B., G.M., and A.D. contributed to data analysis; R.G.-M., M.N.C., and Á.T.K. wrote the manuscript, with all authors correcting the final version.

## Competing interests

The authors declare no competing interests.
