## [Peer Review File · Communications Biology]

Reviewers' comments:

Reviewer #1 (Remarks to the Author):

This manuscript presents a significant amount of work executed carefully. The conclusions are interesting and, in some cases, a little unexpected. In particular, it is interesting that the target transcriptional regulator of the Rap proteins was not predictive of strain fitness. That is, deletions in Rap proteins that target, e.g., Spo0F did not have a generalizable fitness advantage in any conditions. This was also true for the Rap proteins that target other transcription factors. I might have expected those that target exclusively Spo0F, such as RapA and RapB, to have similar fitness advantages.

I wonder whether some of the deletions are having pleiotropic effects due to cross regulation by the deleted phr gene. For example, when rapC-phrC is deleted, it will affect the regulation of all phrC targets such as RapB, RapC, and RapJ. This should be discussed.

I recommend the authors include a schematic depicting the pertinent two-component/phosphorelay pathways and where the Rap proteins and Phr peptides fit into them. Similarly, it would be helpful for the reader if there were tables summarizing the results of the experiments but maybe this is not feasible.

L71 – There are many rap/phr genes whose transcription are uncoupled. I do not think it should be generalized that their transcription is coupled.

L80 – what does “diverse” mean here?

L627 – the initial of the corresponding author’s first name should be capitalized.

L119 – Explain how *B. subtilis* DK1042 is different from NCIB 3610 and refer here to Konkol et al 2013 and/or Nye et al 2017.

L70 – It is mentioned here that rap and phr genes are usually found in pairs in the same genetic loci. While this is true, considering the methodological approach taken in the paper, i.e., deletions of rap-phr cassettes, I think it should be mentioned that Rap proteins can be regulated by Phr peptides that are not encoded by genes in cassettes (e.g., RapB, RapC, and RapJ are controlled by PhrC). And that rap genes sometimes have no nearby or cognate regulatory phr gene.

L172 – “...strains that lack specific Rap-Phr systems show temporal differences in sporulation initiation...” Do the authors expect this to be true for all Rap-Phr pairs or perhaps the Rap proteins that target Spo0F? Please expand on this.

L187 – The references to “a” (A) and “b” (B) panel labels should be capitalized.

L155-156 - ... the initial average population representatives of 1.26%...” If this is calculated simply 100/79 strains then it should be 1.27%

L266 – Please expand on rapX. What is its predicted/known target? Does it have an associated phr?

Reviewer #2 (Remarks to the Author):

The manuscript presents a high-throughput analysis of all the rap/phr gene cassettes of *Bacillus subtilis* and their effect on the fitness of *B. subtilis* under different growth and sporulation regimes. The rap/phr gene cassettes encode peptides that participate in quorum sensing and a regulatory protein that modulates processes such as sporulation and genetic competence in *Bacillus subtilis*. It is still an outstanding question as to what has led to the diversity of the rap/phr cassettes in *B. subtilis* genome, with up to 11 such cassettes being present in some strains. Under the competition experiments used here, the manuscript shows that these rap/phr cassettes are not neutral, and which rap/phr will show an advantage in competition is not predicable. However, I note that I found it difficult to follow the experiment set-up and workflow, and a figure illustrating this would be helpful. Below I indicate a few comments to help clarify the manuscript.

General comments:

1. Provide a figure that illustrates the experiment set-up and workflow. It was difficult to know whether each of the 79 strains were competed together or whether all pair-wise combinations of the individual, double mutants and wild-type were competed.
2. Lines 156-157 indicate that the “majority of experimental replicates showed at least one strain that represented more than 30% of the total population”. It would be helpful to also indicate how variable the results of the technical replicates.
3. Lines 162-163 indicate that there is “very limited time for evolutionary adaptations”, but this statement needs more explanation to know how you arrived at this conclusion.
4. Lines 167-168 and Figure 2 are confusing. Are the results from competition of each mutant against wild-type? You state that figure 2 shows greater variation in the results for the 2-day experiment versus the 5-day experiment. However, in Fig. 2, the 2-day results have larger circles, suggesting more results for a particular rap deletion, which would seem to imply less variation. I may be interpreting this incorrectly, which would indicate that a better explanation is need of this figure.
5. Line 433, you should include information on how the pTB120, pTB233, and pTB234 plasmids were constructed. Supplementary Table 1 simply indicates ‘This Study’ as the reference.
6. Line 477, indicates the number of replicate populations, however, you should also indicate which of these replicates would be considered technical replicates and which would be considered independent, biological replicates.

Reviewer #3 (Remarks to the Author):

The high redundancy of Rap-Phr systems in *B. subtilis* points to their role in the fine-tuning of sporulation or competence kinetics in various environmental conditions. To address this question, the authors used one culture medium (MSgg), two growth conditions (biofilm and planktonic cultures), 78 DNA-bar-coded single and double deletion mutants and different competition experiments, leading to more than 1000 strains being handled. This is a huge work. Yet, it not easy to draw conclusions from this study, first because, as outlined by the authors, of spontaneous mutations leading to genetic drift during the successive reseedings, and second because of the variability within the different mix, of mutant strains outcompeting the others.

Additional questions and comment are listed below:

Supplementary table 1 was unreadable

Was the quorum sensing issue dealt with by the authors ? Phr production by fast-growing strains would affect the Rap activity of slow-growing strain before they reach the threshold density

Is there any orphan Rap in the DK1042 strain ?

Did the authors look for Rap-Phr transfer between the mutants, which could compensate for deletions ?

On what target are the Rap acting to affect the vegetative growth phase ?

How many generations at the end of the experiment, for the 2-days and 5-days conditions, in the pellicle and in liquid cultures ?

Were the sporulation kinetics of mutants compared to the wild-type strain ?

Did the authors check for spores, from the different deletion mutants, resistance to thermal stress ? Slight differences could affect spores survival at 80°C 20 minutes

If 8 out of 10 isolates display mutations, could one expect that up to 80% of the strains have acquired mutations ?

Line 275-279, the sentence is not clear

Line 537, what was the mKate-labelled strain used for ?

Line 566-569, even if the t-test is robust, is the distribution of the log-transformed values close enough to the Gaussian distribution so that this test can be used ?

Reviewers' comments:

Reviewer #1 (Remarks to the Author):

This manuscript presents a significant amount of work executed carefully. The conclusions are interesting and, in some cases, a little unexpected. In particular, it is interesting that the target transcriptional regulator of the Rap proteins was not predictive of strain fitness. That is, deletions in Rap proteins that target, e.g., Spo0F did not have a generalizable fitness advantage in any conditions. This was also true for the Rap proteins that target other transcription factors. I might have expected those that target exclusively Spo0F, such as RapA and RapB, to have similar fitness advantages.

I wonder whether some of the deletions are having pleiotropic effects due to cross regulation by the deleted *phr* gene. For example, when *rapC-phrC* is deleted, it will affect the regulation of all *phrC* targets such as RapB, RapC, and RapJ. This should be discussed.

> In this work, the mega-competition experiment included all the 78 strains at the same time within the same mixture. Therefore, while the *rap-phr* modules are removed in the strains, the other strains in the mixture were still secreting Phr peptides. This suggests that the impact of *phr* deletions is possibly less apparent, but the relevance of *rap* gene deletions is mostly observed. We believe that the newly included figure (Fig. 1) will help to understand this point. We have also included a sentence highlighting that while Phr peptides are available for all the cells, the lack of Rap proteins will impact the cells, i.e., here the lack of response by the Rap protein.

I recommend the authors include a schematic depicting the pertinent two-component/phosphorelay pathways and where the Rap proteins and Phr peptides fit into them. Similarly, it would be helpful for the reader if there were tables summarizing the results of the experiments but maybe this is not feasible.

> We have included a new figure (Fig 1) to provide schemes on (A) the Rap-Phr systems and their tentative targets and (B) on the experimental setup. This should improve easier understanding of the experimental setup as suggested by Reviewers 1 and 2.

L71 – There are many *rap/phr* genes whose transcription are uncoupled. I do not think it should be generalized that their transcription is coupled.

> We have now corrected that the transcription of only some of these *rap-phr* genes are coupled and not general.

L80 – what does “diverse” mean here?

> The term “diverse” has been deleted

L627 – the initial of the corresponding author’s first name should be capitalized.

> The references have been corrected. We apologize for this mistake.

L119 – Explain how *B. subtilis* DK1042 is different from NCIB 3610 and refer here to Konkol et al 2013 and/or Nye et al 2017.

> We have indicated the *comI*^{Q12I} mutation present in DK1042 compared to NCIB 3610 and refer to the original publication Konkol et al 2013.

L70 – It is mentioned here that *rap* and *phr* genes are usually found in pairs in the same genetic loci. While this is true, considering the methodological approach taken in the paper, i.e., deletions of *rap-phr* cassettes, I think it should be mentioned that Rap proteins can be regulated by Phr peptides that are not encoded by genes in cassettes (e.g., RapB, RapC, and RapJ are controlled by PhrC). And that *rap* genes sometimes have no nearby or cognate regulatory *phr* gene.

> **The suggested information has been added to the introduction.**

L172 – “...strains that lack specific Rap-Phr systems show temporal differences in sporulation initiation...”
Do the authors expect this to be true for all Rap-Phr pairs or perhaps the Rap proteins that target Spo0F?
Please expand on this.

> **Indeed, the Reviewer is correct, we have specified this as “Spo0F-specific”.**

L187 – The references to “a” (A) and “b” (B) panel labels should be capitalized.

Corrected.

L155-156 - ... the initial average population representatives of 1.26%...” If this is calculated simply 100/79 strains then it should be 1.27%

> **Indeed, it should be 1.27%. It is now corrected in the manuscript.**

L266 – Please expand on rapX. What is its predicted/known target? Does it have an associated phr?

> **The targets of RapX is unknown (it was just very recently identified in phi3T prophage). We have now included this information.**

Reviewer #2 (Remarks to the Author):

The manuscript presents a high-throughput analysis of all the rap/phr gene cassettes of *Bacillus subtilis* and their effect on the fitness of *B. subtilis* under different growth and sporulation regimes. The rap/phr gene cassettes encode peptides that participate in quorum sensing and a regulatory protein that modulates processes such as sporulation and genetic competence in *Bacillus subtilis*. It is still an outstanding question as to what has led to the diversity of the rap/phr cassettes in *B. subtilis* genome, with up to 11 such cassettes being present in some strains. Under the competition experiments used here, the manuscript shows that these rap/phr cassettes are not neutral, and which rap/phr will show an advantage in competition is not predicable. However, I note that I found it difficult to follow the experiment set-up and workflow, and a figure illustrating this would be helpful. Below I indicate a few comments to help clarify the manuscript.

General comments:

1. Provide a figure that illustrates the experiment set-up and workflow. It was difficult to know whether each of the 79 strains were competed together or whether all pair-wise combinations of the individual, double mutants and wild-type were competed.

> **We have included a new figure (Fig 1) to provide a scheme on (A) the Rap-Phr systems and their tentative targets and (B) on the experimental setup. This should improve easier understanding of the experimental setup.**

2. Lines 156-157 indicate that the “majority of experimental replicates showed at least one strain that represented more than 25% of the total population”. It would be helpful to also indicate how variable the results of the technical replicates.

> **We have included the data on the number of replicates that had at least one strain represented above 25%.**

3. Lines 162-163 indicate that there is “very limited time for evolutionary adaptations”, but this statement needs more explanation to know how you arrived at this conclusion.

> **We agree that this sentence was not supported by data, thus we decided to delete from the manuscript.**

4. Lines 167-168 and Figure 2 are confusing. Are the results from competition of each mutant against wild-type? You state that figure 2 shows greater variation in the results for the 2-day experiment versus the 5-day experiment. However, in Fig. 2, the 2-day results have larger circles, suggesting more results for a particular rap deletion, which would seem to imply less variation. I may be interpreting this incorrectly, which would indicate that a better explanation is need of this figure.

> Figure 3 (previously Fig 2) shows the number of replicates (the size of the bubble) where the given rap-phr mutant has at least or more than 1.27%. Therefore, Figure 3 highlights which mutants had the highest frequency of success in the competition experiment. We have included an extra sentence to make this clearer. Furthermore, we have included an extra Supplementary Figure (2) to make it clearer how many mutants were able to maintain in the population of the replicates. This graph shows that after 2 days of selection (both in pellicles and in planktonic conditions), the number of persisting mutants is higher (several replicates include more than 20 mutant persisting) than the 5 days incubation regimes, where less than 10 mutants could persist at the level or above of the input level. We believe that this additional figure will explain the increased variation (= higher number of mutants being present) in the 2 days incubation conditions.

5. Line 433, you should include information on how the pTB120, pTB233, and pTB234 plasmids were constructed. Supplementary Table 1 simply indicates 'This Study' as the reference.

> We have now indicated in the Methods that the respective antibiotic resistance genes were PCR amplified using oligonucleotides listed in Supplementary table 2.

6. Line 477, indicates the number of replicate populations, however, you should also indicate which of these replicates would be considered technical replicates and which would be considered independent, biological replicates.

> We have used 4 biological and for each 6 technical replicates. This is now stated in the figure legend.

Reviewer #3 (Remarks to the Author):

The high redundancy of Rap-Phr systems in *B. subtilis* points to their role in the fine-tuning of sporulation or competence kinetics in various environmental conditions. To address this question, the authors used one culture medium (MSgg), two growth conditions (biofilm and planktonic cultures), 78 DNA-bar-coded single and double deletion mutants and different competition experiments, leading to more than 1000 strains being handled. This is a huge work. Yet, it not easy to draw conclusions from this study, first because, as outlined by the authors, of spontaneous mutations leading to genetic drift during the successive reseedings, and second because of the variability within the different mix, of mutant strains outcompeting the others.

Additional questions and comment are listed below:

Supplementary table 1 was unreadable

> The supplementary table 1 (text and characters) that includes all strains and plasmids constructed in this study, is properly displayed in our PDF viewers. Yes, it is a long table, nevertheless contains the name and genotype of all strains created here.

Was the quorum sensing issue dealt with by the authors? Phr production by fast-growing strains would affect the Rap activity of slow-growing strain before they reach the threshold density

> Here, we aimed to determine the impact that each Rap-Phr system has on the population fitness of *B. subtilis*, but indeed mostly detected the impact or Rap deletion in our mega-competition experiment, as the mixture of 79 strains will nevertheless produce the quorum sensing molecules. Therefore, the impact on quorum sensing was not examined in this study. We clearly state at the start of the results section

that we were interested in the impact of *rap* module deletion on fitness and how the “different Rap phosphatases would affect the adaptability of *B. subtilis* to different growth conditions”.

Is there any orphan Rap in the DK1042 strain?

> **No orphan Rap is known in DK1042 (or NCIB 3610). The RapP is known to be absent in 168, a domesticated derivative of NCIB 3610, but the pBS plasmid carrying the *rapP* gene has been also detected in numerous other natural isolates (see for example the natural isolates *B. subtilis* MB8_B7, P8_B1 and P8_B1 described in Kiesewalter et al 2020 Microbiol Resour Announc 9: e01406-19).**

Did the authors look for Rap-Phr transfer between the mutants, which could compensate for deletions?

> **We have re-sequenced the genomes of 10 isolates taken after transfer 9 and confirmed the presence of *rap-phr* module deletions in all cases as expected. We have used this approach to verify and confirm the lack of *rap-phr* deletion, thus to be able to claim which mutant was re-isolated additionally to identifying the barcode. This has been now indicated in the genome resequencing paragraph of the results.**

On what target are the Rap acting to affect the vegetative growth phase?

> **We report here the growth properties of *B. subtilis* after 16 hours, thus when cells reach already the stationary phase. It is expected that in case a Rap protein acts on a global regulator that impact the vegetative growth (or simply alters the length of vegetative growth phase due to premature activation of stationery phase processes), mutation in the Rap protein might impact vegetative growth. However, there is no Rap connected *B. subtilis* literature study that has addressed this question.**

How many generations at the end of the experiment, for the 2-days and 5-days conditions, in the pellicle and in liquid cultures?

> **Based on the calculation of 20x dilution at each step, a minimum of 39 generations were present after the 9th cultivation transfers. This information has now been included in the Methods.**

Were the sporulation kinetics of mutants compared to the wild-type strain?

> **There has been no study published in the *B. subtilis* literature that have examined the sporulation kinetics of all mutants in the same genetic background. As stated in the manuscript, our aim was not to examine their particular sporulation kinetics of each mutant or double mutant, but to determine their fitness in a mega-competition containing the mixture of all constructed mutants.**

Did the authors check for spores, from the different deletion mutants, resistance to thermal stress? Slight differences could affect spores survival at 80°C 20 minutes

> **We are not aware of any study in the literature that would have studied the spore resistance properties of *rap-phr* deletions, but the studies so far concentrated majorly on the timing of sporulation initiation.**

If 8 out of 10 isolates display mutations, could one expect that up to 80% of the strains have acquired mutations?

> **We have supplemented the given sentence to be clear that additional mutations were observed in all isolates, but we could identify mutations in 8 out of 10 isolates specifically related to regulatory networks, i.e. “eight out of ten isolated evolved strains had gained mutations in one or more genes involved in the regulatory network controlling population heterogeneity (Supplementary Fig. 1) beside additional mutations”**

Line 275-279, the sentence is not clear

> **We agree that this sentence was overly complicated; therefore, it has been now divided into two sentences and rewritten for easy understanding.**

Line 537, what was the mKate-labelled strain used for?

> The WT strain was labelled with mKATE to facilitate counting of cells on the plate after competition. This information has been added to the materials and methods.

Line 566-569, even if the t-test is robust, is the distribution of the log-transformed values close enough to the Gaussian distribution so that this test can be used?

> Prior to using T-tests, normality was tested by Shapiro-Wilks test. We have now included this information in the Statistical analysis paragraph.

Reviewers' comments:

Reviewer #1 (Remarks to the Author):

The authors have adequately addressed my comments.

Reviewer #2 (Remarks to the Author):

I appreciate the authors detailed responses to the reviewers' comments. In particular the addition of Figure 1 improved the ability of the readers to understand the experimental set up and the system under study. The manuscript's high-throughput analysis of all the rap/phr gene cassettes of *Bacillus subtilis* and their effect on the fitness of *B. subtilis* under different growth and sporulation regimes significantly adds to our understanding of how these quorum-sensing circuits contribute to fitness and evolution.

Reviewer #3 (Remarks to the Author):

Insertion of figure 1, and additional explanations or precisions throughout the manuscript have substantially improved its readability. It remains anyway difficult to understand the reason why the outcome of the competition experiment (which is the main part of this work) is not the same for the different mix. For example in Fig.2, in the 5d pellicle T9 experiment, the EJ mutant is highly successful in mix D but not in mix A or B, and there are other examples of this variability.

Line 102, maybe private and public, since spores are more like private goods

Lines 173-178, the sentence could be divided into two, after 1.27%

Lines 183-184, I still do not understand how, independently from acquired mutations, a slight initial advantage would be lost in the following reseedings

Line 188, replace excel by exceed

Line 196, '... only early spores of most strains ...', or perhaps spores from mutants able to sporulate earlier

Line 272, supplementary File 1 instead of supplementary Fig. 1

Line 324, '...can have an impact on...' would be better than '...have an impact on...' since only some mutations have an impact. All single or double mutants with RapJ have no impact or a negative impact.

Line 377, replace 'the' by 'that' in '... rap-phr mutants the increased their population...'

Answers to reviewers

Reviewer #1 (Remarks to the Author):

The authors have adequately addressed my comments.

> ***We thank the reviewer for this and previous comments.***

Reviewer #2 (Remarks to the Author):

I appreciate the authors detailed responses to the reviewers' comments. In particular the addition of Figure 1 improved the ability of the readers to understand the experimental set up and the system under study. The manuscript's high-throughput analysis of all the rap/phr gene cassettes of *Bacillus subtilis* and their effect on the fitness of *B. subtilis* under different growth and sporulation regimes significantly adds to our understanding of how these quorum-sensing circuits contribute to fitness and evolution.

> ***We thank the reviewer for this and previous comments.***

Reviewer #3 (Remarks to the Author):

Insertion of figure 1, and additional explanations or precisions throughout the manuscript have substantially improved its readability. It remains anyway difficult to understand the reason why the outcome of the competition experiment (which is the main part of this work) is not the same for the different mix. For example in Fig.2, in the 5d pellicle T9 experiment, the EJ mutant is highly successful in mix D but not in mix A or B, and there are other examples of this variability.

> ***We thank the reviewer for thoroughly going through our revision and highlighting these points that we have now fully addressed. We are glad that this Reviewer also acknowledges the improvement in the manuscript. Indeed, there is variability which particular mutant is benefitted in a certain mix. This is the reason why we have use 4 different overnight cultures for each inoculation in addition to being performed in two different days (as described in Methods).***

Strong parallelism might originate from the presence of preexisting mutations in the culture used for inoculation (again, we used four independent overnight cultures for inoculation). The lack of same strain (EJ) being successful in all mixes suggests that this is not general trend. The variability in selection for given strain is discussed in numerous positions in the manuscript (lines 337-345, 358-362, 367-371). Further to get a better overview of the trend, i.e. which mutants are preferably selected, we have presented Figure 3 that helps interpreting the trend within the variability (i.e. quantifying the variability). However, we agree with the reviewer that this specific mutant on mix D was not mentioned, thus now the possibility of preexisting mutation in the inoculation mixture is now mentioned (lines 168-170).

Line 102, maybe private and public, since spores are more like private goods

Lines 173-178, the sentence could be divided into two, after 1.27%

Lines 183-184, I still do not understand how, independently from acquired mutations, a slight initial advantage would be lost in the following reseeding

> Thank you for this comment, important to be clear on this aspect. The abundance in the first cycle depends on fitness, which under planktonic conditions is highly related to growth rate, while under static conditions, the DNA from the first cycle depends on fitness which is affected by the ability to incorporate into the biofilm. Whereas, for the next cycles, the abundance highly depends on sporulation in the previous cycle. This can explain why a slight initial advantage in the first transfer is lost in the following reseedings. For example, a strain that sporulates late will divide and grow to many cells (high abundance in first transfer), compared to one that sporulates early (low in first transfer). However, the one that sporulates early may have produced more spores and will be selected for and thus show higher abundance in the following transfers. This has now been described in the indicated section (line 177-179).

Line 188, replace excel by exceed

Line 196, '... only early spores of most strains ...', or perhaps spores from mutants able to sporulate earlier

Line 272, supplementary File 1 instead of supplementary Fig. 1

Line 324, '...can have an impact on...' would be better than '...have an impact on...' since only some mutations have an impact. All single or double mutants with RapJ have no impact or a negative impact.

Line 377, replace 'the' by 'that' in '... rap-phr mutants the increased their population...'

> We have addressed all these points as suggested by the reviewer. We thank the reviewer for doing a thorough correction in the revised version.

REVIEWERS' COMMENTS:

Reviewer #3 (Remarks to the Author):

The correlation analysis on the variability between mixes at T2 vs T9 does not answer the question of why some mutants are successful in some mixes and not in others. My concern was how this could impact the conclusion on the role of Rap-PhRs in the competition experiments. However, I recognize that this problem is difficult to solve in the frame of the available results, without additional experiments. Given that the authors have answered all the other requests, and that the manuscript nevertheless brings quite interesting data on Rap-PhRs and their involvement in the bacterium adaptation, I will not ask for further changes.

Answer to Editorial comment:

“we ask again that you provide an analysis that shows that the variability between mixes is not a significant contributor to the outcome of the competition experiments and/or that the variability is not greater than the variability due to treatment or genotype, for example, with a PCA that shows samples clustering by genotype/treatment rather than mix.”

We have included a PCA on data aggregated by replicate means showing that data points did not cluster according to mix. We further supported this data by PERMANOVA showing high explanatory power for genotype ($r^2=0.393$, $P=0.001$), whilst mix, in contrast, had none ($r^2=0.00$, $P=1.00$). Overall, this suggests that the major contributor to the observed variation is the genotype rather than the mix.